# Development of a New Trapping System with Potential Implementation as a Tool for Mosquito-Borne Arbovirus Surveillance

**DOI:** 10.3390/insects16060637

**Published:** 2025-06-17

**Authors:** Luísa Maria Inácio da Silva, Larissa Krokovsky, Rafaela Cassiano Matos, Gabriel da Luz Wallau, Marcelo Henrique Santos Paiva

**Affiliations:** 1Departamento de Entomologia, Instituto Aggeu Magalhães (IAM), Fundação Oswaldo Cruz (FIOCRUZ), Av. Professor Moraes Rego, s/n, Campus da UFPE, Cidade Universitária, Recife 50740-465, Brazil; 2Faculty of Mathematics & Science, Brock University, 1812 Sir Isaac Brock Way, St. Catharines, ON L2S 3A1, Canada; 3Programa de Pós-Graduação em Entomologia, Universidade Federal Rural de Pernambuco, Rua Dom Manoel de Medeiros, s/n–Dois Irmãos, Recife 52171-900, Brazil; 4Núcleo de Bioinformática (NBI), Instituto Aggeu Magalhães (IAM), Fundação Oswaldo Cruz (FIOCRUZ), Recife 50670-420, Brazil; 5Department of Arbovirology and Entomology, Bernhard Nocht Institute for Tropical Medicine, D-20359 Hamburg, Germany; 6Aggeu Magalhães Institute (IAM), Universidade Federal de Santa Maria (UFSM), Santa Maria 97105-900, Brazil; 7Núcleo de Ciências da Vida, Universidade Federal de Pernambuco (UFPE), Centro Acadêmico do Agreste-Rodovia BR-104, km 59-Nova Caruaru, Caruaru 55002-970, Brazil

**Keywords:** arbovirus surveillance, Culicidae, trap, saliva, virus RNA

## Abstract

Mosquitoes of the *Aedes* and *Culex* genera are vectors of viruses that cause several diseases in humans. Surveillance of these mosquitoes is important because we can detect viruses in mosquitoes before the disease reaches humans or before epidemics occur. In this work, we adapted a trap that can capture and preserve viral genetic material deposited by mosquitoes when they are attracted to feed. The results from laboratory experiments showed that the trap is capable of preserving this genetic material in different environmental conditions, temperatures, and mosquito densities. The evidence suggests that the trap can be used in the field to support arbovirus surveillance.

## 1. Background

Mosquitoes from the genera *Aedes* and *Culex* are recognized as vectors for several arboviruses, including dengue (DENV), Zika (ZIKV), chikungunya (CHIKV), yellow fever (YFV), Oropouche (OROV), Mayaro (MAYV), and West Nile (WNV) [1,2]. These viruses were responsible for approximately 11 million cases in 2024 in just the Americas, causing disabilities, sequelae, and death [3,4,5,6,7,8,9]. For most of these arboviruses, there are no available vaccines or specific pharmacological treatment, which makes prevention indispensable [10,11].

One fundamental keystone in preventing human infections caused by arboviruses is to continuously survey vectors to early detect virus circulation before human cases occur [12,13]. As the majority of such viruses are constituted by an RNA genome, entomo-virological surveillance depends on maintaining a cold chain from the field to the laboratory, to preserve RNA integrity. This is particularly challenging in most scenarios, especially in remote areas, far from laboratory/research facilities, and in resource-limited settings, in low-income countries and regions. Ensuring proper cold chain management under such conditions is logistically demanding due to the limited infrastructure and personnel [12]. It is worth noting that, although maintaining a cold chain is still recommended, reagents such as *RNAlater*^®^ have been shown to yield comparable results for viral detection [14]. Additionally, viral RNA has been successfully detected by qRT-PCR in dead mosquitoes stored for several weeks under hot and humid conditions. However, qRT-PCR has limitations, as it can only detect RNA from viruses targeted by the specific primers and probes used [12]. In addition, the frequency of mosquito populations infected with pathogenic viruses varies according to transmission periods, often being lower during interepidemic phases, requiring the collection of a large number of specimens for detection, which makes the entire process not cost-effective [2].

Mosquitoes, in general, obtain their energy by feeding on sugar and blood sources throughout their life cycle [15]. Blood feeding is essential for egg production in females after mating, while sugar feeding provides energy for flight, survival, and reproduction [2,15,16]. Female mosquitoes often emerge with low energy reserves and must refill by feeding on plant fluids; however, as females age, and mate, their propensity toward host-seeking behavior increases [16,17]. When a female mosquito feeds on the blood of an arbovirus-infected host, it ingests infectious virions and, if competent, these particles will replicate, disseminate and reach the salivary glands [18,19,20]. In a subsequent blood meal, the virus is inoculated into the vertebrate host through the mosquito’s saliva [21]. During this stage, gravid mosquitoes and females that recently oviposited actively seek sugar to restore their energy status, feeding at the sugar source while expelling infectious saliva [16,22,23].

This biological characteristic makes mosquitoes’ saliva a target sample to screen arbovirus transmission. Flinders Technology Associates^®^ cards (FTA cards; GE Healthcare Life Sciences, Pittsburgh, PA, USA) are a type of filter paper containing chemical compounds and free radical scavengers capable of storing genetic material (including RNA) at room temperature for up to 28 days [24,25]. FTA cards have been employed in the field to capture mosquitoes’ saliva when soaked with honey solution that attracts female mosquitoes seeking an energy source [24,26,27,28]. Different studies have succeeded in adapting FTA cards to fit into mosquito traps, circumventing the need for a cold chain to preserve the virus RNA genomic material [28,29,30]. However, the majority of the traps require electric power, or other resources such as CO_2_ cylinders, which impose additional obstacles for low-income regions and large territories.

Therefore, we adapted and tested, in a laboratory setting, the capacity of a new trap, that does not require electrical power, to capture and preserve infected mosquito saliva at room temperature, coupling to it two FTA cards to benchmark a new tool for arbovirus surveillance in the field.

## 2. Materials and Methods

### BR-ArboTrap Description

BR-ArboTrap was designed to capture and preserve the genetic material obtained from the saliva of mosquitoes that are attracted to a recipient for oviposition. BR-ArboTrap is an adaptation of a trap already used in the field, mainly in Brazil, called Double BR-Ovt, capable of attracting mosquitoes of the species *Aedes aegypti* and *Culex quinquefasciatus* [31].

The BR-ArboTrap (Figure 1 and Appendix A) is composed of a black polyethylene box (35 × 24 × 13 cm) with a central rectangular opening (16 × 9 cm) on the top. A black plastic container for oviposition (15 × 2.5 cm) is placed inside the box, wrapped in a strip of raw cotton (52 × 10 cm), serving as a substrate for the collection of eggs of *Aedes* spp. When setting the trap, the container is filled with water, where 2 g of *Bacillus thuringiensis sorovar Israelensis* (Bti) VectoBac^®^ WG (Tokyo, Japan) (37.4% Bti crystals and spores) is added. Furthermore, two FTA systems are placed laterally in relation to the oviposition container. The FTA system consists of a plastic container with an opening at the top (3.5 cm) to fit the FTA card and sponge. The sponge is soaked in 2 mL of solution containing distilled water, Manuka Honey (Manuka Health^TM^, Auckland, New Zealand) (1 mL each) and two drops of blue food coloring. The sponge is placed in contact with the FTA card and attached to the plastic container so that it faces the opening in the lid, allowing mosquitoes to access the card (Figure 1).

## 3. Laboratory Trap Testing

### 3.1. Artificial Blood Feeding of Mosquitoes

To evaluate the ability of the BR-ArboTrap to attract mosquitoes and preserve genetic material obtained from mosquito saliva, artificial infection assays were conducted using CHIKV-exposed *Ae. aegypti* under laboratory conditions. This combination was selected due to well-documented vector competence, a previous known extrinsic incubation period (average 7–10 days post-exposure), high infection rates, and its use in previous FTA card studies to standardize performance testing [2,32,33].

Mosquitoes from the RecL colony, a well-established colony from the Entomology laboratory at the Aggeu Magalhães Institute in Recife, Brazil, were used in the artificial blood-feeding assays. These mosquitoes were maintained at standard conditions of 26 ± 2 °C, 65–85% relative humidity, and a 10/14 light/dark cycle [34].

Artificial blood-feeding assays were conducted following the protocol described by Krokovsky and collaborators [35]. The number of mosquitoes varied depending on the specific experimental condition (see below); in all cases, an excess of individuals was included to account for mortality and non-fed specimens, ensuring an adequate number of successfully engorged females. Prior to feeding, female mosquitos aged seven-to-ten days were sorted into cages along with half of that number of males. They were maintained with water and 10% sucrose solution ad libitum. To stimulate host-seeking behavior, the sugar solution was withdrawn 24 h before exposure to the blood meal. The mosquitoes were fed using a custom-built artificial feeding device, consisting of a Petri dish with a central opening for introducing the blood mixture. The bottom of the dish was sealed with four layers of Parafilm^®^ (Merck, Darmstadt, Germany), allowing mosquitoes to access the blood. A thermal pack was placed at the top of the dish to maintain the blood mixture at 37 °C; it was replaced once during the one-hour feeding period.

The infectious blood meal was prepared using the CHIKV BRPE408 virus strain (GenBank: MH000700.1), originally isolated from human serum in 2016. For viral propagation, Vero CCL-81 cells (African green monkey kidney, *Cercopithecus aethiops*) were infected with a CHIKV stock solution at a concentration at 1 × 10^8^ PFUs/mL (plaque-forming units per milliliter), using a multiplicity of infection (MOI) of 0.1. The final blood meal was adjusted to a concentration of 3 × 10^4^ PFUs/mL (mean Cq = 16.74, SD = 0.372, mean quantification = 2.5 × 10^13^, Appendix A). Defibrinated rabbit blood, supplied and quality-controlled by the Entomology Department’s insectary at FIOCRUZ, was used in all feeding experiments.

After the feeding period, the mosquitoes were anesthetized on ice for three minutes to facilitate the selection of engorged females, which were subsequently transferred to individual containers. The containers were then housed in insectary cages to maintain biosafety.

All procedures, including artificial feeding and subsequent handling, were performed in duplicates under Biosafety Level 2 (BSL-2) conditions at the infectory of the Entomology Department, FIOCRUZ-IAM.

### 3.2. BR-ArboTrap Experiments

To test the entire trap system and the capacity of the FTA cards to preserve the viral genetic material, a first experiment was conducted, and engorged female mosquitoes were separated into four cages, each one containing approximately 40 individuals (Figure 2A). After seven days post-exposure (dpe), mosquitoes from each cage were released in the containment cage with BR-ArboTrap. After 48 h, FTA cards from each trap were removed and exposed to different environmental conditions to simulate situations in the field or storage. The conditions tested were as follows: 37 °C, 25 °C, −80 °C, and 25 °C with 2 mL of ultrapure water, all for 72 h. Mosquitoes from each trap were aspirated and separated for storage at −80 °C.

To assess the card’s conservation capacity, 1 μL of pGEM^®^-T Easy (50 ng/μL, Promega, Madison, WI, USA) was added as a positive control before mosquito exposure. This approach enabled the differentiation between two scenarios: (i) cards that failed to preserve genetic material, indicated by negative results for both pGEM^®^-T Easy and CHIKV; (ii) cards where CHIKV was not expectorated by mosquitoes, evidenced by a positive result for pGEM^®^-T Easy but a negative result for CHIKV. pGEM^®^-T Easy is a plasmid vector used in cloning, and it was chosen as it is an artificial element that could be distinguished from any mosquito-related molecule.

A second experiment was conducted to test the minimum number of positive mosquitoes that can shed CHIKV RNA to the FTA card (Figure 2B). The mosquitoes were separated into two groups for blood feeding: the CHIKV-exposed group containing approximately 50 females; and the control group containing approximately 30 females. The test group was exposed to CHIKV as mentioned before, and the control group was fed with blood and cell mixture without CHIKV in the same conditions and time as the exposed one.

Following blood feeding, the mosquitoes were sorted into groups of two, five, ten, and twenty individuals per cage. After seven dpe, an FTA system (without the plastic container) was placed on the top of each cage for a one-hour exposure period. The control group remained in the same cage throughout the entire period and was also offered the FTA card system at seven dpe under the same conditions. The retrieved cards were then stored at −80 °C for post-analysis, as well as the females, as mentioned before. The experiment was repeated once under the same conditions.

To compare the quantity of virus released by the mosquitoes in cards and blood, approximately 30 females were exposed to an infected CHIKV blood meal, following the protocol described above, as demonstrated in Figure 2C. After exposure to the infective meal, the specimens were moved into 20 small cages, each one containing one individual. At seven dpe, two different exposure systems were offered to separate groups of 10 mosquitoes each: one group was provided with an FTA card system (without the plastic container), while the other received 2 mL of the defibrinated rabbit blood in a Petri dish, as previously described. After one hour, the FTA cards were collected, residual blood was removed by pipetting, and the mosquitoes were aspirated. All samples were then stored at −80 °C for subsequent analysis. The experiment was repeated once under the same conditions.

### 3.3. Mosquitoes and FTA^®^ Cards’ Processing

Mosquito females aspirated from the traps were visually screened under a stereomicroscope to assess feeding status by detecting the presence of the blue coloration on the abdomen. Each specimen was macerated in 300 μL of UltraPure™ DNase/RNase-Free Distilled Water (Invitrogen™, Waltham, MA, USA) using sterilized pistils and automatic homogenizer. FTA^®^ cards were processed following Ritchie et al.’s (2013) protocol [26]. Each pair of FTA cards collected from the trap was aseptically cut into strips over a Petri dish using sterile scissors and forceps, and then transferred to a 1.5 mL microcentrifuge tube containing 500 μL of ultra-pure water. The tubes were kept on ice and subjected to vortexing cycles (10 s every 4 min) for a total duration of 20 min. Subsequently, the liquid content was transferred to a sterile 5 mL syringe and filtered into a new 1.5 mL tube for downstream processing. The pair of FTA cards from the entire trap experiment (Figure 2A) was processed as a pool.

For RNA extraction, 100 μL of each sample was processed using a modified TRIzol™ Reagent (Invitrogen™, Waltham, MA, USA) protocol as follows [36]. Briefly, 100 μL of tissue homogenate was mixed with 200 μL of TRIzol reagent and vortexed for 15 s, followed by a 5 min incubation at room temperature. Next, 100 μL of chloroform was added, and the mixture was manually agitated for 15 s. After an additional 2–3 min of incubation at room temperature, the samples were centrifuged at 12,000× *g* for 15 min at 4 °C. The aqueous phase was carefully transferred to a new tube containing 250 μL of 100% isopropanol, gently mixed, and incubated at room temperature for 10 min. A second centrifugation at 12,000× *g* for 10 min at 4 °C was performed. The supernatant was discarded, and the resulting RNA pellet was washed with 300 μL of 75% ethanol, gently mixed, and centrifuged at 7500× *g* for 5 min at 4 °C. After removing the ethanol, the RNA pellet was air-dried for approximately 15 min and subsequently resuspended in 30 μL of RNase-free water.

DNA extraction in samples from the first experiment (Figure 1A) was performed following DNAzol™ manufacturer protocol (Invitrogen™, Waltham, MA, USA) using 100 μL of processed FTA. PCR was conducted for the amplification of the M13 region from the pGem^®^-T Easy vector using the PCR Master Mix amplification kit (Promega, Madison, WI, USA) and primers M13F and M13R [37]. Electrophoresis was performed in a 3% agarose gel in 0.5X TBE, stained with 5 μL of GelRed^®^ (10,000×) (Biotium^TM^, Fremont, CA, USA), subjected to 120 volts for one hour, and visualized on an ultraviolet transilluminator.

### 3.4. Viral Detection

CHIKV Reverse transcriptase quantitative PCR (RT-qPCR) assays were performed for each extracted RNA sample, using QuantiNova Probe RT-PCR Kit (Qiagen, Hilden, Germany) following the conditions described in the work of Krokovsky et al. (2022) [38], and primers and probes for this reaction were described by Lanciotti et al. (2007) (Appendix A) [39]. The samples were tested in duplicates, using a negative template control (all reagents except RNA), a negative extraction control, and positives (standard curve) were also included. A standard curve was generated following the method described by Kong et al. (2006), with modifications [40]. It consisted of six dilution points, each containing a known number of RNA molecules. Details of standard curve construction are described in Appendix A. The reactions were performed using a QuantStudio 5 Real-Time PCR System (Applied BioSystems, Waltham, MA, USA) and the results were analyzed in QuantStudio™ Design and Analysis Software v1.5 with absolute quantification of the number of the RNA copies based on the standard curves, and with the automatic threshold and baseline. Samples with Cq (Cycle quantification) values ≤ 38.5 in duplicates were considered positive. Details of sensitivity are described in Appendix A.

### 3.5. Statistical Analysis

GraphPad Prism V. 8 software (https://www.graphpad.com/) was used to assess all statistical analyses and graphs. Ordinary One-Way ANOVA was used for all experiments using the Multiple Comparisons tool, and for calculating the mean value with a 95% confidence interval (CI). For the analyses, duplicates were considered and included, and the minimum and maximum values of the RNA copy number/mL axis were automatically specified by the software. To evaluate the samples’ positivity, we considered negative samples as follows: samples with mean Cq of ≥38.5 or samples annotated as undetermined by RT-qPCR (values that do not exceed the Cq threshold) [41]. Both categories were included in the ANOVA calculations.

## 4. Results

To evaluate the capacity of BR-ArboTrap to capture and store viral RNA material from arbovirus-exposed mosquitoes, we conducted artificial infection assays with *Ae. aegypti* and CHIKV. After the artificial blood feeding, 96% of females were positive with CHIKV in seven dpe, with a mean number of copies of CHIKV RNA/mL of 1 × 10^13^ (ranging from 1 × 10^12^ to 5 × 10^13^) (Table 1, Appendix A). Considering all experiments, 90% of FTA cards exposed to positive females were positive for CHIKV, with a mean number of RNA copies/mL of 2 × 10^9^ (ranging from 4 × 10^7^ to 7 × 10^9^) (Table 1, Appendix A).

When analyzing FTA cards’ capacity to store CHIKV RNA after exposure to different temperatures (Figure 3A), neither water nor temperature (25 °C, −80 °C, 37 °C) impacted the mean number of CHIKV RNA copies (*p* = 0.5008, Appendix A). All FTA cards tested positive for the virus, with a mean CHIKV RNA copy number of 2 × 10^9^ (Table 1, Figure 3A). Regarding females, 114 out of 120 were positive for CHIKV, with 1 × 10^13^ mean number of CHIKV RNA copies (Table 1). However, a significant *p*-value was observed when comparing mosquitoes in the 37 °C group, indicating a higher number of CHIKV RNA copies in these samples (*p* < 0.0001, Table 1, Figure 3A, Appendix A).

Regarding the abdomen status of mosquitoes analyzed in the entire trap experiment, of the total of 120 retrieved females, 40 showed blue coloration on the abdomen (33%) (Figure 4A). The amplification of pGEM^®^-T Easy, used as a positive control (Figure 2A), showed an expected fragment size of 250 bp when visualized through agarose gel electrophoresis in all FTA card samples (Figure 4B).

All FTA cards exposed to different groups of two, five, and twenty virus-exposed mosquitoes were positive for CHIKV, with a mean number of CHIKV RNA copies/mL of 2 × 10^9^ (Table 1, Figure 3B, Appendix A). Within the group of ten females, one card exposed to positive females was negative (mean number of CHIKV RNA copies/mL of 4 × 10^7^ and mean Cq > 38.5). There was no significant difference in the value of CHIKV RNA copies/mL between the cards from the four groups analyzed (*p* = 0.4739, Appendix A). In this experiment, 67 out of 68 females that were exposed to CHIKV were positive at seven dpe, with an 8 × 10^12^ mean number of CHIKV RNA copies/mL, and no significant difference between the groups was observed (*p* = 0.4739, Table 1, Figure 3B, Appendix A). Six females died during the experiment and were excluded fromthe analysis.

The last experiment was divided as follows: a group of CHIKV-exposed mosquitoes that took the second meal on blood, and a group of CHIKV-exposed mosquitoes that took the second meal on an FTA card (Figure 3C). Six females from each group died during the experiment and were excluded from the analysis. Regarding blood samples exposed to positive mosquitoes, 11 out 14 were positive for CHIKV, with a mean number of RNA copies/mL 9 × 10^7^ (Figure 3D, Table 1, Appendix A). In this first group, of the 14 females exposed to CHIKV, 13 were considered positive, with a 4 × 10^13^ mean number of RNA copies/mL (Figure 3C, Table 1, Appendix A). In the second group, 12 of 14 FTA cards, exposed to females positive for CHIKV, were positive, with mean number of RNA copies of 5 × 10^7^ CHIKV RNA copies/mL (Figure 3D, Table 1, Appendix A). In this group, all females were considered positive. Excluding of the analysis one blood sample that was offered to a negative female, 78% (11/14) of blood positivity versus 85% (12/14) of FTA card positivity was observed, with no significant difference (*p* = 0.782) in the mean number of CHIKV RNA copies/mL between these two groups (Figure 3D, Appendix A).

## 5. Discussion

Entomo-virological surveillance is a fundamental activity in countries affected by arbovirus outbreaks as it can anticipate human infection [12]. However, the requirement of a cold chain to preserve viral RNA poses a significant logistical barrier, preventing the establishment and continuity of actions in the field. In this study, we adapted and tested a novel trap based on FTA technology capable of capturing and preserving arboviruses secreted by mosquitoes without the need for refrigeration or electricity. Our experiments demonstrated that the BR-ArboTrap effectively retains viral RNA, even under adverse environmental conditions, making it a promising tool for enhancing arbovirus surveillance.

Multiple studies have previously conducted laboratory experiments to evaluate the efficacy of FTA cards for capturing and storing mosquito released viruses. Hall-Mendelin, and colleagues [24] demonstrated that honey-soaked FTA cards could effectively bind the Ross River virus (RRV) at 23 °C for at least 28 days. Subsequently, *Ae. aegypti* mosquitoes, artificially infected with WNV, RRV, or CHIKV, showed successful virus detection on the cards, with a 75% positivity rate for CHIKV. However, despite this high detection rate, no blue coloration was observed in mosquito abdomens, suggesting that saliva expectoration occurs regardless of honey ingestion [24]. This could indicate that relying on abdominal dye may underestimate actual expectoration rates in some mosquito species [24].

In contrast, our study found a 90% positivity rate for CHIKV on FTA cards, with 35% of mosquitoes presenting blue abdomens. Unlike previous studies, we tested the entire trap system, which included the oviposition container filled with water and the sponge—elements that may have contributed to higher humidity and better honey conservation, leading to the higher detection rates. As Fourniol and collaborators [32] highlighted, the honey concentration in the feeding solution plays a crucial role, as a higher honey content may enhance saliva expectoration, with a 1:1 ratio yielding the highest viral loads.

Melanson et al. [42] detected DENV-2 in 65% of FTA cards when testing it within an artificially infected mosquitoes’ assay. The virus was detected on cards at dilutions ranging from 0.1 to 1000 plaque-forming units (PFUs)/µL (one PFU average), which allowed for a correlation with Cq values ranging from 40 to 28, respectively [42]. This indicates that one PFU of DENV-2 can be detected on a card through RT-qPCR (with a Cq of 38), which sets FTA virus detection limits [42]. In our study, we observed an average Cq of 36 for cards and a Cq of 17 for mosquitoes, similar to the results reported by Melanson: Cq of 36 and 20, respectively.

To explore the practical application of FTA cards for field deployment, we evaluated their ability to preserve viral RNA under different environmental conditions. We observed no difference in positivity rates when the cards were exposed to water and extreme temperatures, simulating tropical field conditions. Hall-Mendelin and collaborators [24] also tested card storage at −80 °C, finding no significant impact on RNA integrity [25]. However, Krambrich and colleagues [43] examined the impact of exposing FTA cards to 37 °C for extended periods (one to 30 days) and found that viral RNA of tick-borne encephalitis virus and Japanese encephalitis virus became undetectable by RT-qPCR after seven days. This suggests a risk of false-negative results if the cards are stored at temperatures above 37 °C for more than a week [44].

To differentiate between FTA cards that were not exposed to CHIKV-positive mosquito saliva and those yielding false-negative results, and given the absence of a standardized salivary marker, we introduced pGEM^®^-T Easy as a positive control prior to mosquito exposure. This strategy allowed for the identification of two distinct outcomes: (i) cards that failed to preserve genetic material, indicated by negative results for both pGEM^®^-T Easy and CHIKV; and (ii) cards where CHIKV was not expectorated by mosquitoes, reflected by a positive result for pGEM^®^-T Easy but a negative result for CHIKV. pGEM^®^-T Easy was selected as the positive control for two primary reasons: first, as an artificial marker, it is easily distinguishable from any endogenous mosquito or pathogen-derived nucleic acids, ensuring unambiguous detection; second, it avoids the use of RNA that could potentially be re-acquired by mosquitoes during feeding despite FTA cards being known to inactivate viruses and degrade nucleic acids [24,44]. This positive control was applied exclusively in the initial experiment, in which all cards tested positive for pGEM^®^-T Easy and none tested negative for CHIKV, supporting its potential for use in future assays. However, as this control was not incorporated in subsequent experiments, additional studies are needed to validate its consistency and reliability across different experimental conditions.

When comparing positivity rates between FTA cards and blood samples, no substantial differences were observed. In a scenario where each FTA card and blood sample was exposed to a single CHIKV-positive female, the positivity rate was 85.7% for FTA cards and 78.5% for blood samples. Additionally, the mean number of CHIKV RNA copies present in these groups did not differ significantly between the two groups (*p* = 0.782). Our findings align with those of Fourniol and collaborators [32], who tested the efficacy of FTA cards using *Ae. aegypti* artificially infected with CHIKV, finding no significant difference between the quantity of virus expectorated in blood and in FTA card. These results indicate that using FTA cards for saliva collection did not compromise virus detection and may serve as a superior alternative to other methods, such as forced salivation, which is not only highly labor-intensive but also underestimates transmission rates [33].

Collectively, the evidence from previous studies suggests that several factors influence virus detection on FTA cards: mosquito species, the number of infected mosquitoes, the dpe of mosquito feeding, the amount of honey, the duration of mosquito feeding on the card, exposure to extreme temperatures, and storage time [25,32,43,44]. Among these factors, the latter three appear to be the most influential [43]. Despite the uncertainties surrounding these factors, it was possible to detect viruses like CHIKV, WNV, RRV, Barmah Forest virus, Murray Valley encephalitis virus, Usutu virus, Kunjin virus and Allfuy virus on FTA cards deployed in the field [24,28,30,45,46]. These findings are promising for arbovirus surveillance, as similar results may not always be achievable through traditional mosquito collection methods due to RNA degradation or the labor-intensive process of handling large sample volumes.

In this study, we tested, in the laboratory, the entire trap system that can be deployed in the field, providing more realistic insights compared to experiments using only FTA cards. However, this study did not evaluate several other variables that could potentially influence virus detection on FTA cards: mosquito age, different dpe, other mosquito species, sugar processing, and the amount of saliva. A limitation of BR-ArboTrap is that it is not possible to identify the specific mosquito species that deposited the virus; therefore, it is not viable to identify the potential vector.

## 6. Conclusions

In this study, we designed a trap to overcome the challenges of arbovirus surveillance in mosquitoes by circumventing the need for a cold chain and exploiting the biological need of mosquitoes to feed on a sugar source. The trap operates without electrical energy and is easy to install, transport, and maintain in the field, making it suitable for widespread use by field agents. It effectively attracts adult mosquitoes, captures their saliva, and preserves arbovirus-exposed samples for subsequent RNA detection. Additionally, the use of FTA cards ensures sample stability, enabling collection and transportation to remote locations. These features, along with the availability of inexpensive and accessible materials, make the BR-ArboTrap an innovative and adaptable tool for arbovirus surveillance, particularly in low-income regions.

## Figures and Tables

**Figure 1 insects-16-00637-f001:**
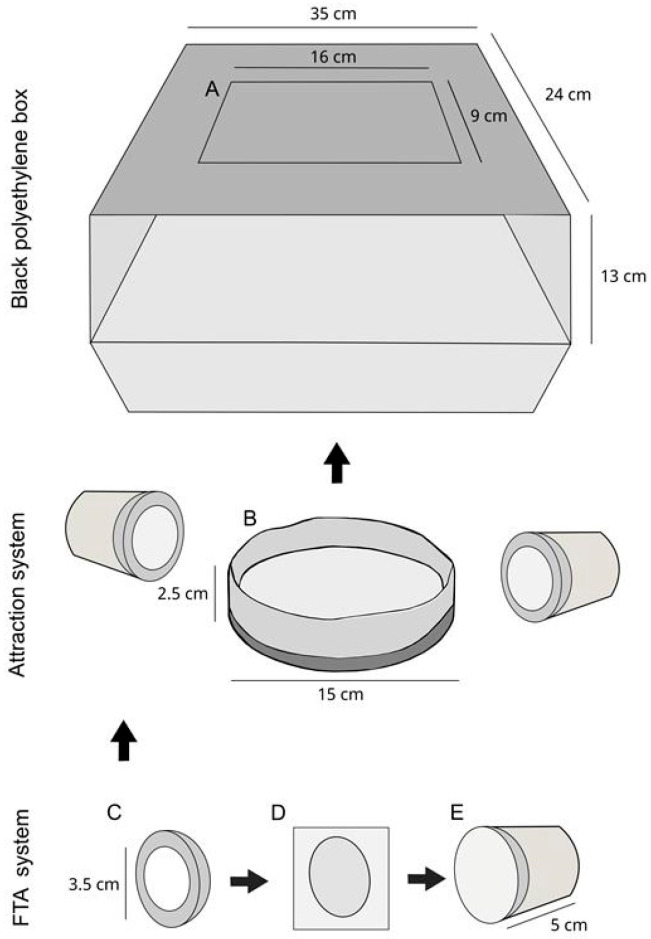
Schematic figure of BR-ArboTrap description showing the trap components. (**A**) Top opening within the black polyethylene box. (**B**) Oviposition container. (**C**) Top-opening jar lid. (**D**) FTA card placed on top of the sponge soaked in the solution. (**E**) Plastic jar used for support for the FTA card and the sponge.

**Figure 2 insects-16-00637-f002:**
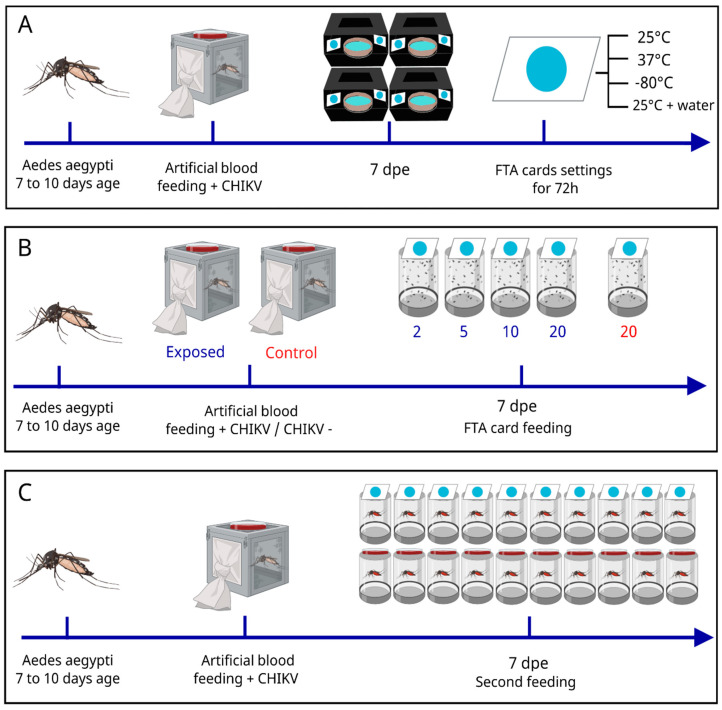
Description of laboratory experiments conducted to evaluate the BR-ArboTrap. Three artificial infection assays were performed using *Aedes aegypti* fed on CHIKV-infected blood. (**A**) Experiment designed to assess the performance of the BR-ArboTrap system and the stability of FTA cards under varying environmental conditions. After feeding, the mosquitoes were distributed among four traps, and at seven dpe, the FTA cards were collected and subjected to four different environmental conditions for 72 h. (**B**) The experiment was executed to determine the minimum number of positive mosquitoes required for virus detection in FTA cards. Two groups of mosquitoes were used: the experimental group was exposed to blood infected with CHIKV, and the control group, with 30 mosquitoes, was exposed to uninfected blood. After feeding, CHIKV-exposed mosquitoes were divided into five groups, with two, five, ten, and twenty mosquitoes per container. (**C**) Comparison of CHIKV RNA copy numbers detected in FTA cards versus blood samples. Mosquitoes were exposed to blood infected with CHIKV and subsequently allocated individually in separate containers. At seven dpe, one group of 10 mosquitoes was offered uninfected blood, while the other group of 10 was provided with an FTA card. Detection of CHIKV was conducted in mosquitoes, FTA card, and blood samples.

**Figure 3 insects-16-00637-f003:**
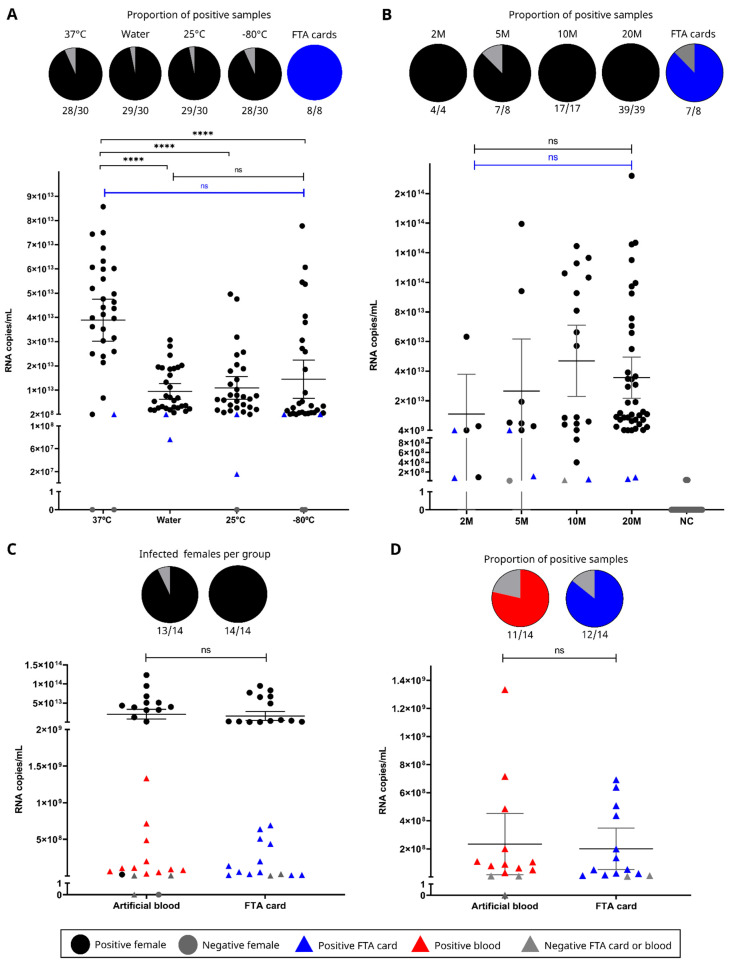
Graphical representation of the results from laboratory experiments using the BR-ArboTrap for chikungunya virus (CHIKV) detection in mosquitoes, FTA cards, and blood samples. (**A**) Positivity of FTA cards following salivation by *Aedes aegypti* mosquitoes exposed to CHIKV, including evaluation of card stability under varying environmental conditions and the proportion of CHIKV-positive females per trap. (**B**) Assessment of the minimum number of CHIKV-exposed mosquitoes (two, five, ten, and twenty) required for CHIKV RNA detection on FTA cards, along with the proportion of CHIKV-positive females per trap. (**C**) Comparison of CHIKV RNA copy numbers detected in FTA cards versus those found in blood samples, accompanied by the proportion of CHIKV-positive females per group. Statistical comparison between the artificial blood group and FTA card group is also shown. (**D**) Magnified view of panel (**C**), highlighting the comparison and statistical significance of CHIKV RNA copy numbers (per mL) between FTA card and blood samples, as well as the respective positivity rates. Grey vertical bars indicate the mean with 95% confidence intervals. Horizontal bars mark groups subjected to statistical analysis. Blue represents FTA card samples; black represents other groups. Statistical notation: ns = not significant; **** = *p* < 0.0001. Pie charts illustrate the number of CHIKV-positive samples for mosquitoes (black), FTA cards (blue), and blood (red). Black circles represent positive females; blue triangles, positive FTA cards; red triangles, positive blood samples; grey shapes indicate negative results. NC = negative control group samples.

**Figure 4 insects-16-00637-f004:**
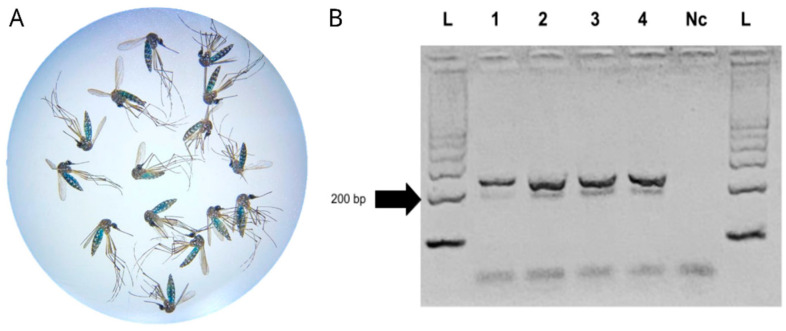
The results of the experiment conducted with the entire trap. (**A**) Females *Ae. aegypti* retrieved from the trap, showing blue abdomen coloration at stereomicroscope; (**B**) agarose gel showing amplification of pGEM^®^-T Easy, used as a positive control for FTA^®^ cards. L, 1 kb molecular weight marker plus DNA Ladder; 1 to 4, PCR products of the M13 region of pGEM^®^-T Easy from traps 1 to 4; Nc, PCR reaction negative control.

**Table 1 insects-16-00637-t001:** RT-qPCR results for CHIKV detection mosquitoes, FTA cards, and blood samples, presenting the mean CHIKV RNA copies/mL, mean Cq, and positivity rates per group across three independent experiments. “Mean” refers to the average of each parameter within a single experiment, while “Total Mean” represents the overall average calculated from all samples across all experiments. Values in parentheses indicate standard deviations. Positivity rate is expressed as the number of positive samples over the total number of samples analyzed. M, mosquito sample; F, FTA sample; B, blood sample; - not tested; * not applicable.

Experiment	Groups	Mean Number of CHIKV RNA Copies/mL per Group			Mean Cq			Positivity		
		M	F	B	M	F	B	M	F	B
1	37 °C	2 × 10^13^	2 × 10^9^	-	15 (±1.23)	35 (±2.43)	-	28/30	2/2	-
	25 °C + Water	4 × 10^12^	7 × 10^9^	-	16 (±2.72)	35 (±1.54)	-	29/30	2/2	-
	25 °C	7 × 10^12^	5 × 10^9^	-	16 (±2.93)	34 (±0.97)	-	29/30	2/2	-
	−80 °C	3 × 10^12^	5 × 10^9^	-	16 (±3.74)	33 (±1.88)	-	28/30	2/2	-
	Mean	1 × 10^13^	2 × 10^9^	-	15 (±2.87)	34 (±1.52)	-	*	*	*
2	2 M	1 × 10^12^	2 × 10^9^	-	23 (±8.8)	37 (±0.7)	-	4/4	2/2	-
	5 M	4 × 10^12^	2 × 10^9^	-	19 (±7.14)	36	-	7/8	1/2	-
	10 M	5 × 10^13^	4 × 10^7^	-	18 (±7.7)	38	-	17/17	2/2	-
	20 M	1 × 10^13^	9 × 10^7^	-	16 (±3.5)	37	-	39/39	2/2	-
	Mean	1 × 10^13^	9 × 10^7^	-	17 (±6.25)	37 (±0.83)	-	*	*	*
3	Second meal on FTA	4 × 10^12^	5 × 10^7^	-	15 (±5.4)	36 (±2)	-	13/14	12/14	-
	Second meal on blood	4 × 10^13^	-	9 × 10^7^	16 (±1.3)	-	36 (±1.7)	14/14	-	11/14
	Mean	3 × 10^13^	5 × 10^7^	9 × 10^7^	16 (±0.3)	36 (±2)	36 (±1.7)	*	*	*
Total mean		1 × 10^13^	2 × 10^9^	9 × 10^7^	16 (±2.5)	36 (±1.5)	36 (±1.7)	*	*	*

## Data Availability

No new data were created or analyzed in this study. Data sharing is not applicable to this article.

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
