# Peer review of "Development of a New Trapping System with Potential Implementation as a Tool for Mosquito-Borne Arbovirus Surveillance"

_insects, 2025, doi:10.3390/insects16060637_

Round 1
Reviewer 1 Report
Comments and Suggestions for Authors
The article "Development of a New Trapping System with Potential Implementation as a Tool for Mosquito-Borne Arbovirus Surveillance" by Inácio da Silva et al. explores the use of an ovi mosquito trap for detecting the Chikungunya virus on FTA cards. This study addresses the necessity for an independent detection method that does not rely on a cold chain for mosquito transport to a laboratory. The concept is noteworthy, given the increasing importance of finding efficient and sensitive methods in response to the rise in arbovirus cases. In my assessment, the study should include at least three replicates for each experimental condition and enhance the statistical analyses. Additionally, it should provide methodological details (see below) and include amplification curves from the rt-qPCR to demonstrate the test's sensitivity, along with details on the quantification of the standards used. Observations on the manuscript: Line 82: Consider indicating that preservatives of genetic material, such as RNA Shield or RNAlater, have also been utilized. Lines 172-175: While the manuscript refers to details of blood feeding, it is crucial for the authors to specify the number of mosquitoes fed, the feeding duration, the blood type used, and the feeding method. Line 22: The term "Incciso D" is unclear. Does it pertain to another figure? Lines 234-235: Ensure consistency with the number or legend regarding the number of mosquitoes. Lines 235-237: Was there an embedded sponge on top of the FTA cards, as previously described, "soaked in 2 mL of solution containing distilled water, Manuka Honey (Manuka Health) (1 mL each)"? Lines 256-260: The authors should describe the details of the genetic material elution, even if a reference is provided. Lines 261-263: The authors must detail any modifications to the extraction protocol. Viral Detection Section: Include cycling conditions and primers in a table. In addition to the standard curves, was a positive control used for the qPCR? It is important to present the Cq and amplification curve of the viral isolate with which the mosquitoes were fed, corresponding to the 3×10^4 PFU/mL. Detail the origin and concentrations used to construct the standard curve, and provide details according to the reference. Show the efficiency and slope of the standard curve. Line 311: The authors are advised to incorporate an image of the standard curve, highlighting some of the amplification curves from the experimental samples. Line 337: What is the rationale for the gray portion in the circle representing 16/16 infected females in Figure 3C? Additionally, the circle for the negative female is absent from the graph on the left side.
Line 351: Does this imply that it was utilized solely as a positive control in FIG 2A? What, then, serves as the control in the other experiments?
Line 367: The current data may suggest an overestimation of quantifications. To mitigate any uncertainties, the authors should provide information from the standard curve and specify the concentrations used. In other absolute quantification systems, a Cq of 38 corresponds to approximately 1x10(1), whereas here, 4x10(7) is mentioned.
Line 387: Why does the number of FTA cards not correspond to the number of mosquitoes in Figure 3C? It is understood that each mosquito was assigned an FTA card, and if four per group have died, there should be 16 mosquitoes remaining. However, why are there not 16 FTA cards?
Line 440-442: Based on this comparison, and as previously noted, it is crucial for the authors to specify the sensitivity of their test, indicating the minimum detectable amount in genomes or PFU.
Line 462-464: This should be clarified in the materials and methods section. The question arises as to why this control was not employed in all experimental conditions. For instance, in the experiment with 10 mosquitoes, only one of the two FTA cards is positive. Without the positive control, how should this data be interpreted?
Line 486-487: In a pool trial, such as the one involving 10 mosquitoes on an FTA card, the number of infected mosquitoes in that pool may significantly influence the outcome, potentially explaining why one of the two FTAs is not positive.
Line 502-504: The purpose of the device containing eggs is unclear. Is it intended solely to attract gravid females, or is it also used to obtain larvae for subsequent species identification? Please clarify this point.
Figure 1: It is recommended that the authors include a photograph of the trap to aid in its description.
Figure 3A: The authors should adjust the "y" axis to allow visualization of the negative FTA cards at the level of 0 copies of RNA.
Table 1: Insert a line between the experiments to enhance reader comprehension.
Author Response
Reviewer 1:
The article "Development of a New Trapping System with Potential Implementation as a Tool for Mosquito-Borne Arbovirus Surveillance" by Inácio da Silva et al. explores the use of an ovi mosquito trap for detecting the Chikungunya virus on FTA cards. This study addresses the necessity for an independent detection method that does not rely on a cold chain for mosquito transport to a laboratory. The concept is noteworthy, given the increasing importance of finding efficient and sensitive methods in response to the rise in arbovirus cases. In my assessment, the study should include at least three replicates for each experimental condition and enhance the statistical analyses. Additionally, it should provide methodological details (see below) and include amplification curves from the rt-qPCR to demonstrate the test's sensitivity, along with details on the quantification of the standards used.
Observations on the manuscript:
Line 82: Consider indicating that preservatives of genetic material, such as RNA Shield or RNAlater, have also been utilized.
Thank you for your comment. We have added a statement noting that RNAlater® can be used as an alternative for preserving samples containing viral RNA.
Lines 172-175: While the manuscript refers to details of blood feeding, it is crucial for the authors to specify the number of mosquitoes fed, the feeding duration, the blood type used, and the feeding method.
Thank you for your comment. The procedural details have been provided in the Methods section.
Line 22: The term "Incciso D" is unclear. Does it pertain to another figure?
Thank you for the observation. Figure 2D does not exist; the legend in question was intended to refer to Figure 3D. We have corrected the figure reference accordingly.
Lines 234-235: Ensure consistency with the number or legend regarding the number of mosquitoes.
Thank you. This issue has been addressed. The numbers have been corrected, and a clarification has been added to the Methods section indicating that a surplus of mosquitoes was initially fed to ensure the target number post-feeding, accounting for mortality and unfed individuals. The final values have been adjusted accordingly.
Lines 235-237: Was there an embedded sponge on top of the FTA cards, as previously described, "soaked in 2 mL of solution containing distilled water, Manuka Honey (Manuka Health) (1 mL each)"?
Yes, this refers to what we described as the FTA system without the plastic container, which consists of the FTA card and a sponge soaked with the solution. This is already detailed in the following sentence: “An FTA system (without the plastic container) was made available for one hour on the top of the cage. The control group remained in the same cage for seven days, and then the card was submitted to the same process.” To improve clarity, we have revised the second sentence to read: “…and then the FTA card system was also offered.”
Lines 256-260: The authors should describe the details of the genetic material elution, even if a reference is provided.
Thank you for your comment. The procedural details have been provided in the Methods section.
Lines 261-263: The authors must detail any modifications to the extraction protocol.
Thank you for the observation. We have incorporated the modification of the protocol into the relevant section accordingly.
Viral Detection Section: Include cycling conditions and primers in a table. In addition to the standard curves, was a positive control used for the qPCR? It is important to present the Cq and amplification curve of the viral isolate with which the mosquitoes were fed, corresponding to the 3×10^4 PFU/mL. Detail the origin and concentrations used to construct the standard curve, and provide details according to the reference. Show the efficiency and slope of the standard curve.
Thank you for your thoughtful comments. We agree that including this information enhances the strength of our manuscript. Regarding the cycling conditions and the sequences of primers and probes, all relevant details have been added to Supplementary Tables S1 and S2. A separate positive control was not employed, as the standard curve was constructed using RNA extracted from the same CHIKV BRPE408 viral stock used in the artificial feeding assays. As noted, the origin of the standard curve has now been specified in the main text, with detailed procedures provided in Supplementary Section 1. Additionally, we have included the amplification plots, quantification data, and parameters values for the standard curve in Supplementary Figure S1 (panels A and B) and Table S3, S4. Supplementary Figure S1 (panels C and D) also displays the amplification plot and standard curve corresponding to the viral quantification used in the feeding assays. The mean Cq value was 16.74 (SD = 0.372), and the mean viral load was 2.5 × 10¹³ copies of CHIKV RNA/mL. All of this information has been incorporated into the main text accordingly.Supplementary Figure S1 (panels E and F) show the amplification plot and and standard curve of a plate containing samples tested.
Line 311: The authors are advised to incorporate an image of the standard curve, highlighting some of the amplification curves from the experimental samples.
Thank you for your comment. We agree that this information will significantly strengthen our results. We have now included both details as Supplementary Figure S1.
Line 337: What is the rationale for the gray portion in the circle representing 16/16 infected females in Figure 3C? Additionally, the circle for the negative female is absent from the graph on the left side.
Thank you for this observation. Upon identifying this and other inconsistencies, we conducted a thorough review of all data and updated the relevant information throughout the main text, tables, and supplementary materials. We apologize for the oversight and have taken the necessary steps to ensure accuracy.
Line 351: Does this imply that it was utilized solely as a positive control in FIG 2A? What, then, serves as the control in the other experiments?
pGEM®-T Easy was designed as a positive control for scenarios in which an FTA card might yield entirely negative results. However, its performance was uncertain, as it had not previously been used as a control in conjunction with FTA cards. Since the experiments were conducted in parallel, we opted to evaluate its applicability only in the first experiment (Figure 2A).
Line 367: The current data may suggest an overestimation of quantifications. To mitigate any uncertainties, the authors should provide information from the standard curve and specify the concentrations used. In other absolute quantification systems, a Cq of 38 corresponds to approximately 1x10(1), whereas here, 4x10(7) is mentioned.
Thank you for your comment and for the opportunity to clarify the use of our standards. The standards were prepared and quantified as described in the Viral Detection section of the Methods and in Supplementary Section 1, as also noted in a previous response. The concentration of the stock used to generate the standard curve dilutions was 350 ng/μL, which corresponds to approximately 4 × 10¹² RNA copies/mL, as calculated using the following formula.
Number of RNA copies = (X g/μL RNA) / (N × 340) × 6.02 × 10²³
Where:
X = amount of RNA (g/ μL)
N = amplicon length
6.02 × 10²³= Avogadro’s constant
The primer set used in this study was originally described by Lanciotti et al. (2007), who reported a sensitivity threshold of 0.9 PFU. However, this value is expressed in a unit that is not directly comparable to our results, which are quantified in RNA copy numbers. Furthermore, the conditions under which their assay was conducted—including equipment, reaction volumes, and reagent concentrations—differ substantially from those used in our study. Our analysis of the standard curve revealed that the lowest dilution point yielding consistent and reproducible amplification had a Cq value of 35.049, corresponding to approximately 4.9 × 10⁷ RNA copies. Therefore, we defined our positivity threshold as a Cq value of 38.5, corresponding to approximately 4 × 10⁶ RNA copies/mL.
We acknowledge the concern, as some RT-qPCR protocols described in the literature report lower detection limits (as low as 100–10 copies). However, our assay was specifically optimized for our sample matrix and experimental cost constraints. Based on our validation, we are confident in the accuracy and reliability of the quantification and sensitivity achieved. All relevant data from the sensitivity assay are provided in the supplementary material Figure S2 and tables S3 and S4 and are referenced in the Viral Detection section of the manuscript.
Line 387: Why does the number of FTA cards not correspond to the number of mosquitoes in Figure 3C? It is understood that each mosquito was assigned an FTA card, and if four per group have died, there should be 16 mosquitoes remaining. However, why are there not 16 FTA cards?
Thank you for pointing this out. You are correct—an inconsistency was identified due to the unintentional duplication of one FTA card sample, resulting in an unequal number of samples between groups, with one additional FTA card sample compared to blood samples. To ensure consistency and enable valid comparisons between groups, the extra FTA card sample has been excluded. We have carefully reviewed all data, including those from Experiments 1 and 2, and re-performed the statistical analyses accordingly. The necessary corrections have been applied to the graph, the main text, the relevant table, and the supplementary materials.
Line 440-442: Based on this comparison, and as previously noted, it is crucial for the authors to specify the sensitivity of their test, indicating the minimum detectable amount in genomes or PFU.
Thank you for your comment, we addressed this question in the comment above.
Line 462-464: This should be clarified in the materials and methods section. The question arises as to why this control was not employed in all experimental conditions. For instance, in the experiment with 10 mosquitoes, only one of the two FTA cards is positive. Without the positive control, how should this data be interpreted?
Yes, as mentioned in a previous comment, pGEM®-T Easy was designed as a positive control for scenarios in which an FTA card might yield completely negative results. However, we were uncertain about its performance, as it had not previously been used as a control in conjunction with FTA cards. Since the experiments were conducted in parallel, we opted to test this control only in the first experiment (Figure 2A). Given this limited application, we cannot confirm whether pGEM®-T Easy would perform similarly under different experimental conditions. Its use in this context serves as a preliminary exploration, and therefore we cannot state with certainty that it fully addresses the concern raised.
It is worth noting that similar findings have been reported by other researchers using FTA cards under field conditions. For example, Hall-Mendelin et al. (2010) observed that mosquitoes may have expectorated virus-containing saliva during the infectious phase, but by the time of card processing, the insects were no longer infectious—possibly accounting for negative FTA card results. In our study, as you mentioned in a subsequent comment, "the number of infected mosquitoes in a given pool may significantly influence the outcome, potentially explaining why one of the two FTA cards tested negative." This is a plausible explanation, along with other variables such as viral load and the volume of saliva expectorated.
REFERENCES:
Hall-Mendelin, S., Ritchie, S. A., Johansen, C. A., Zborowski, P., Cortis, G., Dandridge, S., Hall, R. A., & van den Hurk, A. F. (2010). Exploiting mosquito sugar feeding to detect mosquito-borne pathogens. Proceedings of the National Academy of Sciences of the United States of America, 107(25), 11255–11259. https://doi.org/10.1073/pnas.1002040107
Line 486-487: In a pool trial, such as the one involving 10 mosquitoes on an FTA card, the number of infected mosquitoes in that pool may significantly influence the outcome, potentially explaining why one of the two FTAs is not positive.
Yes, we addressed this comment in the anterior question.
Line 502-504: The purpose of the device containing eggs is unclear. Is it intended solely to attract gravid females, or is it also used to obtain larvae for subsequent species identification? Please clarify this point.
The oviposition container is a key component of the trap, specifically designed to attract gravid females seeking suitable sites for egg-laying. As these females have already taken a blood meal—a necessary step for egg maturation—they are more likely to harbor pathogens. Additionally, the oviposition container serves as a tool for identifying the mosquito species attracted to the trap. However, it is important to emphasize that a direct correlation cannot be established between the mosquitoes that interacted with the FTA card and those that deposited eggs. This is because a mosquito may be drawn exclusively to the FTA card without visiting the oviposition container, or conversely, may deposit eggs without coming into contact with the card.
Figure 1: It is recommended that the authors include a photograph of the trap to aid in its description.
Thank you for the insight. We added the photography of BR-ArboTrap in Supplementary Figure 1.
Figure 3A: The authors should adjust the "y" axis to allow visualization of the negative FTA cards at the level of 0 copies of RNA.
Thank you for the observation. We have made the necessary adjustments to ensure that negative samples are now visible in the graphs. To accommodate all data points—including negative values—we modified the Y-axis scale individually for each graph. As a result, the Y-axis is no longer standardized across graphs, which may affect direct visual comparisons between them.
Table 1: Insert a line between the experiments to enhance reader comprehension.
Line added.
Reviewer 2 Report
Comments and Suggestions for Authors
In this present manuscript, the authors described a novel mosquito trap with potential application for mosquito-borne arbovirus surveillance in the field under versatile conditions. The purpose of the trap design was to conserve high quality RNA sample from mosquito saliva for lab testing. The methodology was validated in the lab, however, there are several aspects that the authors need to address to improve the value of their studies,
Major comments:
- The BR-ArboTrap was a combination of FTA cards and a Double BR-Ovt system. It may improve mosquito attraction compared to the traditional FTA card and not require electricity and CO2, but, in theory, should not improve RNA sample storage and RNA preservation. The authors should address how BR-ArboTrap outdoes other methods regarding RNA preservation for downstream analysis.
- Multiple conditions were tested and compared in a lab setting. Was there any data to support the feasibility of the BR-ArboTrap application in the field? How well can the honey solution attract mosquitoes in the field? The authors should also elaborate on the potential challenges and limitations of this system.
- The authors should reorganize the manuscript- balance the Method, Result, and Discussion sections. The readability of this current manuscript is low. For example, line 348-354, the authors should address the purpose of the pGEM-T vector feeding experiment and the meaning of the results.
- The authors should proofread the manuscript and pay attention to details. For example, 1/ unit is missing from line 311 to line 315; 2/ Figure 3A has lower error bars but not the other panels; 3/ line 222, Figure 1D is missing.
- In Table 1, please explain the meaning of Mean values and Total mean values. How to interpret them? The author should also present the data from negative control groups in all experiments (mosquitoes fed on diets without CHIKV).
Author Response
Reviewer 2:
In this present manuscript, the authors described a novel mosquito trap with potential application for mosquito-borne arbovirus surveillance in the field under versatile conditions. The purpose of the trap design was to conserve high quality RNA sample from mosquito saliva for lab testing. The methodology was validated in the lab, however, there are several aspects that the authors need to address to improve the value of their studies,
Major comments:
The BR-ArboTrap was a combination of FTA cards and a Double BR-Ovt system. It may improve mosquito attraction compared to the traditional FTA card and not require electricity and CO2, but, in theory, should not improve RNA sample storage and RNA preservation. The authors should address how BR-ArboTrap outdoes other methods regarding RNA preservation for downstream analysis.
We appreciate the reviewer’s thoughtful observation and agree that the primary aim of the BR-ArboTrap is not to enhance RNA preservation per se. Rather, its strength lies in circumventing the need to preserve whole mosquitoes by capturing and stabilizing viral RNA directly onto FTA cards in the field. This approach significantly reduces logistical burdens associated with sample storage and transport—especially in remote areas without access to refrigeration or specific reagents—while maintaining RNA integrity sufficient for downstream molecular analyses. Thus, the innovation of the BR-ArboTrap resides not in outperforming existing methods in RNA preservation capacity, but in offering a practical, cost-effective, and electricity-independent alternative for reliable arbovirus surveillance.
Multiple conditions were tested and compared in a lab setting. Was there any data to support the feasibility of the BR-ArboTrap application in the field? How well can the honey solution attract mosquitoes in the field? The authors should also elaborate on the potential challenges and limitations of this system.
We are currently organizing the data from the field deployment of the BR-ArboTrap for future publication, which will provide detailed insights into its feasibility and performance under real-world conditions. The effectiveness of honey as an attractant has been well-documented in previous studies cited in our manuscript, including Wipf et al. (2019), Birnberg et al. (2020), and Fynmore et al. (2021). These studies demonstrated that honey-baited FTA cards effectively attract mosquitoes in diverse environments, supporting their use in arbovirus surveillance. Additionally, we have addressed the potential challenges and limitations of the BR-ArboTrap system in the final paragraph of the Discussion section. One primary limitation is that the trap does not retain the mosquitoes, and therefore cannot directly identify the species depositing saliva onto the FTA card. However, in the event of a positive viral detection in the field, follow-up entomological investigations can be carried out to determine the likely vector species.
Despite this limitation, our laboratory results demonstrate that the BR-ArboTrap effectively attracted mosquitoes, which successfully deposited saliva onto the FTA cards. These cards subsequently tested positive for viral RNA, confirming the trap's potential utility for arbovirus detection.
REFERENCES:
Wipf, N. C.,et. al (2019). Evaluation of honey-baited FTA cards in combination with different mosquito traps in an area of low arbovirus prevalence. Parasites & vectors, 12(1), 554. https://doi.org/10.1186/s13071-019-3798-8;
Birnberg, L.; Temmam, S.; Aranda, C.; Correa-Fiz, F.; Talavera, S.; Bigot, T.; Eloit, M.; Busquets, N. Viromics on Honey-Baited FTA Cards as a New Tool for the Detection of Circulating Viruses in Mosquitoes. Viruses 2020, 12, 274. https://doi.org/10.3390/v12030274; Fynmore, N., Lühken, R., Maisch, H. et al. Rapid assessment of West Nile virus circulation in a German zoo based on honey-baited FTA cards in combination with box gravid traps. Parasites Vectors 14, 449 (2021). https://doi.org/10.1186/s13071-021-04951-8;
The authors should reorganize the manuscript- balance the Method, Result, and Discussion sections. The readability of this current manuscript is low. For example, line 348-354, the authors should address the purpose of the pGEM-T vector feeding experiment and the meaning of the results.
We thank the reviewer for this valuable suggestion. In response, we have clarified the rationale for using the pGEM-T Easy vector as a positive control in the Methods section and have expanded the Discussion to better contextualize its role and inherent limitations within our experimental framework. These revisions aim to improve clarity and ensure that the purpose and interpretation of this experiment are more transparent to the reader. We trust that these adjustments contribute to enhancing the manuscript’s overall readability and balance across sections.
The authors should proofread the manuscript and pay attention to details. For example, 1/ unit is missing from line 311 to line 315; 2/ Figure 3A has lower error bars but not the other panels; 3/ line 222, Figure 1D is missing.
Thank you for your observations. We have added the missing unit (CHIKV RNA copies/mL) at the indicated location and reviewed the terminology throughout the manuscript to ensure consistency. Regarding what we believe was referred to as Figure 2D—also noted by another reviewer—there was a labeling error: Figure 2D did not exist, and the legend intended for Figure 3D was mistakenly placed under Figure 2. This mislabeling has now been corrected. We also addressed the issue of missing error bars. This occurred because, in GraphPad Prism software, certain values could not be displayed when the Y-axis was set to a logarithmic scale. After changing the Y-axis to a linear scale—as requested to allow for the inclusion of negative values—the error bars are now correctly displayed.
In Table 1, please explain the meaning of Mean values and Total mean values. How to interpret them? The author should also present the data from negative control groups in all experiments (mosquitoes fed on diets without CHIKV).
We thank the reviewer for this important observation. To clarify, the term Meanrefers to the average value for each category calculated from all samples within a single experiment, whereas Total Meanrepresents the overall average across all experiments for each category. We acknowledge that the previous terminology was unclear and have now revised the figure legend to explicitly define these terms. We also identified and corrected an issue where summed positivity rates were mistakenly labeled as "Mean" and "Total Mean"—a misrepresentation that could lead to confusion. These aggregated values have been removed and replaced with "* – not applicable" to ensure accurate interpretation.Regarding the inclusion of negative control data, we fully agree with the reviewer’s point. While we chose to present these data in Figure 3 for improved visual clarity—given that most values were zero—we emphasize that all individual results remain accessible in Additional File 3: Table S5. We hope these revisions collectively enhance the manuscript’s clarity and transparency.
Round 2
Reviewer 1 Report
Comments and Suggestions for Authors
The authors have followed all the previous recommendations, being more precise in the methodology and the presentation of the results, so I consider this work can be accepted in its current form.
Author Response
Comment: The authors have followed all the previous recommendations, being more precise in the methodology and the presentation of the results, so I consider this work can be accepted in its current form.
Response: Thank you for all the suggestions.
Reviewer 2 Report
Comments and Suggestions for Authors
The quality was improved in this revised manuscript, and the data presented are now more convincing. However, the nature of this study is for better data collection in the field, which means that without any field trial, it is hard to support the innovation of BR-ArboTrap. I will suggest including at least some pilot studies in the field under different conditions, and including more comparisons with previous models, such as the Double BR-Ovt system.
Author Response
Comment: The quality was improved in this revised manuscript, and the data presented are now more convincing. However, the nature of this study is for better data collection in the field, which means that without any field trial, it is hard to support the innovation of BR-ArboTrap. I will suggest including at least some pilot studies in the field under different conditions, and including more comparisons with previous models, such as the Double BR-Ovt system.
Response: We appreciate the reviewer’s thoughtful feedback and recognition of the improvements made to the manuscript. However, we respectfully clarify that the inclusion of field trials or pilot studies is beyond the scope of the current work. Our study was specifically designed to develop and validate the BR-ArboTrap under controlled laboratory conditions, as a necessary preliminary step prior to field deployment. Field trials require additional time, ethical approvals, inter-institutional agreements, and logistical coordination with public health authorities, and therefore constitute a separate research effort for a future study.
Additionally, we emphasize that the BR-ArboTrap and the Double BR-Ovt system serve fundamentally different purposes: the former is designed to collect mosquito saliva for virological surveillance, while the latter targets oviposition and adult mosquito capture. As such, direct comparisons between these tools are not methodologically appropriate, as they are intended for distinct surveillance strategies. We hope this clarification is helpful and appreciate the opportunity to improve the clarity of our manuscript.